# Enhancing Food Safety through Adoption of Long-Term Technical Advisory, Financial, and Storage Support Services in Maize Growing Areas of East Africa

**Samuel K. Mutiga [1,2,*]**, **Arnold A. Mushongi [3]** and **Erastus K. Kangéthe [4]**

1   Biosciences Eastern and Central Africa-International Livestock Research Institute (BecA-ILRI) Hub, ILRI Complex, Along Old Naivasha Road, Uthiru Market, PO Box 30709-, GPO Nairobi 00100, Kenya
2   Department of Plant Pathology, University of Arkansas, 2217-A Plant Science Building, Fayetteville, AR 72701, USA
3   Tanzanian Agricultural Research Institute (TARI)-Ilonga, P.O Box 33 Kilosa, Morogoro 67409, Tanzania; amushongi@gmail.com
4   PO Box 34405-, Nairobi 00100, Kenya; mburiajudith@gmail.com
*   Correspondence: skm88@cornell.edu

**Abstract:** Grain production and storage are major components in food security. In the ancient times, food security was achieved through gathering of fruits, grains, herbs, tubers, and roots from the forests by individual households. Advancements in human civilization led to domestication of crops and a need to save food for not only a household, but the nation. This extended need for food security led to establishment of national reservoirs for major produces and this practice varies greatly in different states. Each of the applied food production, handling, and storage approaches has its benefits and challenges. In sub-Saharan Africa, several countries have a public funded budget to subsidize production costs, to buy grains from farmers, and to store the produce for a specific period and/or until the next harvests. During the times of famine, the stored grains are later sold at subsidized prices or are given for free to the starving citizens. If there is no famine, the grain is sold to retailers and/or processors (e.g., millers) who later sell it to the consumers. This approach works well if the produce (mainly grain) is stored under conditions that do not favor growth of molds, as some of these microbes could contaminate the grain with toxic and carcinogenic metabolites called mycotoxins. Conditions that alleviate contamination of grains are required during production, handling, and storage. Most of the grain is produced by smallholder farmers under sub-optimal conditions, making it vulnerable to colonization and contamination by toxigenic fungi. Further, the grain is stored in silos at large masses, where it is hard to monitor the conditions at different points of these facilities, and hence, it becomes vulnerable to additional contamination. Production and storage of grain under conditions that favor mycotoxins poses major food health and safety risks to humans and livestock who consume it. This concept paper focuses on how establishment of a local grain production and banking system (LGPBS) could enhance food security and safety in East Africa. The concept of LGPBS provides an extension of advisory and finance support within warehouse receipt system to enhance grain production under optimal conditions. The major practices at the LGPBS and how each could contribute to food security and safety are discussed. While the concept paper gives more strength on maize production and safety, similar practices could be applied to enhance safety of other grains in the same LGPBS.

**Keywords:** maize; food safety; community-based support systems; integrated mycotoxin control strategies

## 1. Introduction

Maize is a staple food to the East African people and is consumed at a per capita of slightly over 90 kg per annum in the region [1]. Because of the dietary value attached to maize in Kenya and Tanzania, a lack of maize is synonymous to a "famine hit". To cushion the consumers against famine, the governments provide subsidized fertilizers and buy maize grains and store them at national strategic grain reserves, which are located at the major producing regions [2–4]. The quality and quantity of the produce stored in the strategic reserve depends on farm management and the immediate post-harvest handling techniques [5]. Cultivation of maize under stress induced by abiotic or biotic conditions can lead to low yield and poor-quality grain. Furthermore, inadequate application of farm inputs would affect the quality and quantity of the grain [6]. In East Africa, the majority of maize is produced by small-scale farmers under sub-optimal farm conditions [2,7]. Maize that is produced under drought and under low nitrogen was reported to have high chances of contamination by mycotoxins, toxic, and carcinogenic substances produced by some grain molds [8,9]. To enhance the quality of harvested maize, there is a need to adopt strategies that can enable farmers to produce the crop under the optimal conditions. Because the optimal maize production conductions are diverse and costly to achieve, there is need to provide financial and advisory services to the small-scale maize growers. Here, we propose a concept through which the farmers could be given long-term supports to achieve the optimal maize production conditions through community-based non-governmental business institutions. The proposed concept seeks to overcome challenges stemming from lack of farm inputs, expensive farm labor services, and lack of knowledge on proper farm practices.

Although different countries have tried to subsidize fertilizer, other maize stress factors such as drought, weeds, pests and diseases, and post-harvest handling methods affect the quality of the grain [9,10]. Additional deterioration in grain quality occurs when the produce is stored under conditions that favor the growth of toxigenic molds and toxin production [11]. To ensure safety of grain in the national strategic reserve, workers of the reserve facility conduct grain moisture content tests upon delivery of the grain by the farmers. If maize has moisture of above 13%, the farmers are required to dry the grain to the recommended content, mainly by pouring and spreading it on the surface of plain polythene sheets, which are placed on the ground under the sun [12]. Once dried, the grain is repackaged into gunny bags and reweighed before it is purchased for storage within the facility. The number of bags stored within a facility depends on the annual government budget and the availability of the commodity in the respective regions during the specific season. The storage facilities consist of huge silos where the gunny bags are kept in large masses. These bags are stored for 4–12 months. In Kenya, massive mycotoxin contamination has been reported in farmers' storage sheds and the national maize reserves [13]. Application of better grain handling and storage methods can reduce loss of maize quality and mycotoxin contamination [14]. Unfortunately, most of the maize growers do not have the capacity to overcome the predisposing conditions for mycotoxin contamination in maize. The grain is produced, handled, and stored under sub-optimal conditions that favor colonization by toxigenic molds.

Availability of food does not fully address the food security concerns if the food is unsafe and/or is contaminated. While Africa struggles to boost its food production to feed the burgeoning population, food safety challenges arise from several environmental contaminants such as industrial wastes, sewage systems, plant toxins, food additives, pesticide residues, and mycotoxins. At farm level, fungal colonization and subsequent contamination by mycotoxins in food stuff occurs produced, handled and/or stored under conditions that favor growth of molds and toxigenesis [14]. Mycotoxins contribute to the huge burden of food contaminants across the world and pose a particularly large health risk to maize consumers in East Africa [13,15]. The chronic risk factor for mycotoxins is higher than for any of the other contaminants [16]. There is an increasing concern that extreme conditions due to climate change could worsen food safety in Africa [17]. Efforts to mitigate mycotoxins, particularly aflatoxin contamination in East African maize, have recently gained momentum, but there is still need for more action. Given the complexity of the problem, there is a need

to adopt concerted approaches to effectively prevent factors that favor contamination in the entire maize value chain (Figure 1).

Mycotoxins of importance in the East African maize value chain include aflatoxin and fumonisin [15,18,19]. Aflatoxin is mainly produced by *Aspergillus flavus,* a maize pathogen which causes ear rot, and by *A. nomius* and *A. parasiticus* [13]. Aflatoxin B1 is the most potent carcinogen known globally and has been reported to contaminate maize and peanuts in many parts of Eastern and Central Africa [20]. The factors for and the effects of the toxin are described below, as it represents a classical case for the complexity of mycotoxin contamination in maize value chain (Figure 1). Acute exposure to aflatoxin routinely causes lethal poisoning, including recent outbreaks in Kenya and Tanzania [21,22]. Chronic exposure to aflatoxin has been associated with multiple health issues such as liver cancer, reduced nutrient absorption, stunting of children, poor fetal development, immunosuppression, and a general increase in morbidity [23,24]. When livestock are fed with feed that is contaminated by AFB1, the toxin is metabolized to release another potentially harmful substance called aflatoxin M1 (AFM1). This metabolized aflatoxin has been reported in livestock products such as eggs, milk, and cheese [25–27]. Higher chances of liver cancer were observed in hepatitis B patients whose urine tested positive for the AFM1 [25]. Another major maize contamination is fumonisin and is produced by Fusarium fungal species. Fumonisin is a carcinogen and is widespread in maize-growing areas of East Africa, particularly Tanzania and Kenya [15,19,28].

There is an urgent need to establish effective and sustainable mycotoxin mitigation strategies because the toxins cause many known socio-economic and health impacts in African livelihoods [14,29]. Mycotoxins affect trade because contaminated grains and peanuts from Africa cannot be accepted in markets of countries with stringent regulatory systems [30]. Furthermore, exposure to mycotoxins affects the health of livestock and humans [30]. Exposure to mycotoxins in livestock occurs through ingestion of contaminated feeds [27]. Exposure of humans occurs through ingestion of contaminated foods derived from cereals such as maize or peanuts, and livestock products such as milk and eggs [27]. Acute exposure is fatal and occurs when high doses of the toxins are consumed. For example, in 2004, 125 people died after consumption of maize that was highly contaminated with aflatoxin in Kenya [21]. Chronic exposure to mycotoxins is the gradual ingestion of small doses over long time. Chronic exposure has been reported to cause perpetual hidden human health problems in Africa [23,25]. Studies have shown that many mycotoxins have carcinogenic and antinutritional factors. Mycotoxins are thought to alter the cellular and biochemical functions of the intestine, resulting in micronutrient deficiencies, systemic immune activation, and impaired nutrient uptake [31].

A strong association between poverty and aflatoxin exposure was reported in a cross-sectional study conducted in Kenya [32]. Based on urine and albumin tests, people from a lower economic class were found to be more likely to be exposed to aflatoxin. People with low income have less access to basic items, including clean water, balanced diets (particularly dietary diversity, as this reduces chances of mycotoxin exposure), and are likely to have multiple health problems (e.g., depressed body immunity). Mycotoxin exposure is more harmful to people with compromised body immune systems, and it can lead to perpetual health risks [29]. Further studies found that malnourished children who suffered from kwashiorkor and had been exposed to aflatoxin had a higher morbidity, and were found to have lower hemoglobin, edema, increased infections, and spent longer durations in hospitals [33]. The increased morbidity and reduced nutrient absorption, due to the antinutritional characteristics of mycotoxins, could be the major cause of the reported strong association between exposure to mycotoxins and stunting in children, which has been reported across several countries in Africa [31]. Additionally, hepatitis B patients who had been exposed to aflatoxin were found to be more likely to develop liver cancer [1,34]. Notwithstanding the economic losses caused by impaired maize trade due to mycotoxin, the implications of the health burden in exposed societies is immense. Many studies provide associations between chronic exposure to mycotoxins and perpetual sickness in society, a condition that results in physical and mental weakness, depression, inability to work, and hence,

low income [1,31,33]. A society filled with depressed people becomes a home for crimes (e.g., riots, suicides, thefts, etc.) [35].

Based on available literature for aflatoxin (as this is a globally known case) contamination in African food and feed value chains, we synthesized the complex relationship between the driving factors and the recognized/perceived outcomes. Results of this synthesis are summarized into an aflatoxin problem tree, whose roots are the factors that drive, and branches are the outcomes of the contamination and exposure of humans and livestock (Figure 1). The factors that predispose maize to colonization by toxigenic fungi and subsequent contamination by aflatoxin include the biology of the resident fungi (abundance and toxigenicity potential), maize (host resistance), environmental stress on maize crop (drought, inadequate/excess soil nutrients, and damage by pests), and improper handling of grain during and after harvesting [36,37]. Generally, maize stress leads to colonization and contamination by mycotoxins [8,37]. Once the grain is colonized, its level of contamination begins in the field and could continue if stored under conditions that favor aflatoxin formation [19].

The ideal strategies to effectively manage aflatoxin and fumonisin contamination would be to prevent plant stress during crop production in the field, ensure proper handling during harvesting, and to store the grain under conditions that do not favor growth and contamination by the toxigenic molds. Unfortunately, such ideal conditions are not easy to achieve by all stakeholders in the value chain. Many of the small-scale holders are faced with challenges of meeting the high costs of maize production inputs. Further, farmers may not have the capacity to grow and handle the grain under conditions that prevent peri- and post-harvest contamination [38]. Although the majority of the small-scale grain traders sort or blend to reduce the concentration of apparent moldiness in the purchased grain, this practice has not been proven to significantly reduce aflatoxin, but fumonisin [15]. Additionally, if the grain is sold to traders at high moisture content, the level of contamination could increase during storage in a wait for the sale to the millers and the grain reserve. In Kenya and Tanzania, it is customary for farmers and traders to spend weeks in a queue of trucks for the grain to be offloaded at the millers and at the grain reserves. This delay in offloading of trucks could be a terrible cause of deterioration and mycotoxin contamination in grain that might have been harvested at a moisture content above 13%. Upon delivery at the storage silos, grain that is already colonized by toxigenic molds is likely to have more contamination, if the conditions are not checked.

In the current setting, the East African national grain reserves serve the storage purpose. However, establishment of facilities that can tackle other factors affecting maize value chain, alongside the storage and safety concerns, is still prudent. For the theme of enhancing maize productivity and safety, there is a need to establish sustainable systems that not only address soil fertility and grain storage, but also can minimize other risks for mycotoxin contamination at pre-, peri-, and post-harvest stages. To deal with the challenges arising from the national produce and produce board (NCPB), Kenya has passed a policy to create a warehouse receipt system (WRS) for grains [39]. The WRS allows farmers or traders to deposit their grain at a nearby certified warehouse facility and then be issued with a document of title called a warehouse receipt. The farmer or trader can then apply for short-term credit from a participating bank or other financial institution using the warehouse receipt as security for a loan, thus increasing access to finance for small-scale farmers [40]. While this is a good idea, the design implies that the credit accessed would differ based on the income of the farmer or trader. However, because harvested grain is consumed by the whole community and country, at large, there is need to expand this system to ensure that farmers are given enough support to produce to the capacity of their farms. Here, we propose a concept of a community-supported farming system, to exist as an expansion of the WRS through provision of financial credit and crop management advisory system at different stages in the value chain. It is envisaged that implementation of the concept can boost maize production, productivity, profitability, and effectively reduce contamination and human exposure to the damaging toxins.

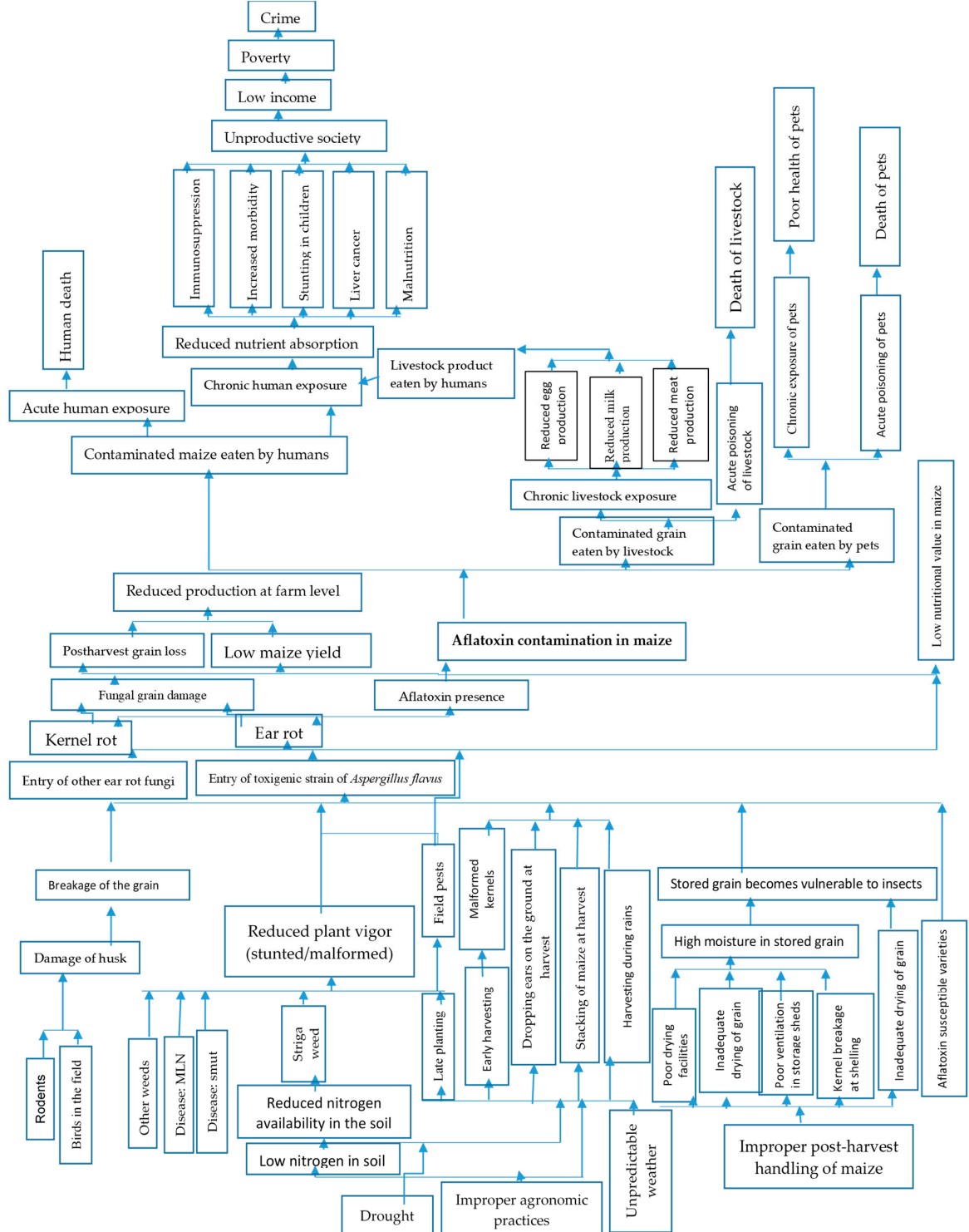

**Figure 1.** The aflatoxin problem tree. Diagrammatic representation of factors and outcomes for aflatoxin contamination in maize.

## 2. The Concept of Local Grain Production and Banking System (LGPBS)

### 2.1. Description of LGPBS

Conceptually, LGPBS refers to centers that provide farm inputs, management practice advisory services, grain aggregation, grain storage, grain drying, grain safety assessment, and credit facilities to farmers in the neighborhoods where the grains are grown. These centers could be established through

collaborations between corporate organizations, local, and/or national governments. The centers could operate within delineated maize-growing areas, which could be termed as maize production schemes. The sizes of the schemes and capacity of the centers would depend on the intensity of grain production within a given area. Individual centers should be able to efficiently provide key support service to the farmers within the respective schemes. LGPBS would ensure that the grain is produced under optimal conditions by providing advice and inputs to farmers to ensure that the crop has minimal stress. They would further participate in provision of facilities that ensure that the harvested grain is appropriately handled and tested prior to storage under the custody of the WRS. Because the centers serve many farmers in a given schemes, they would participate in finding potential grain buyers, and in turn, farmers would deposit the grain. Per the WRS program, farmers can withdraw their grain on a regular basis and can even acquire credit depending on the value of their contribution in the center [40]. Owing to the economies of scale enjoyed by bringing the farmers together, the LGPBS would play a key role in enhancing better livelihoods of the participant farmers. They would serve as key points through which any interested governmental and non-governmental organizations can channel their support for the farmers in the region. The sustainability of the centers would be through the business transactions with the farmers and external resource mobilization e.g., support by local and national governments, development partners, and donors.

Major stakeholders of the LGPBS would comprise the farmers, ministries of agriculture, agricultural research institutions, health and water organizations from local and national governments, private investors, agro-dealers, seed companies, national food safety organizations, microfinance organizations, and insurance companies. Each of these bodies would contribute at different points in the grain value chain (Figure 2). Although the government agencies would have a stake, a preferred model is to have LGPBS operate as independent businesses with autonomous management. The autonomous management would enhance efficiency by eradication of bureaucracy and political interference; hence, better service delivery to the farmers. The key sections of the LGPBS would work in tandem to ensure that all major services are efficiently provided to the farmers, with an overall goal of enhancing increased maize productivity and safety within the respective production schemes.

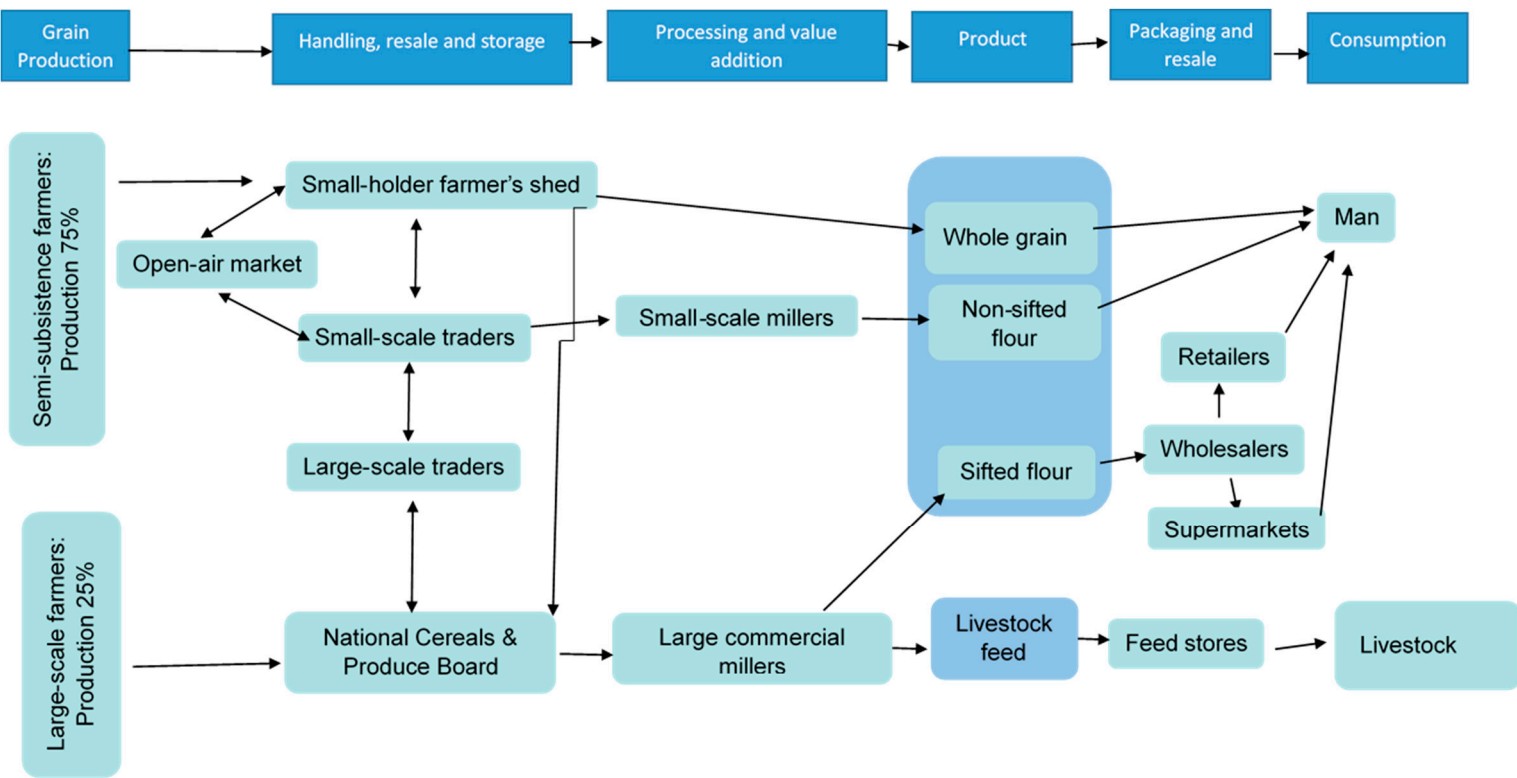

**Figure 2.** A schematic summary of East African maize value chain showing the importance of small-scale growers. The small-scale growers are faced with multiple production constraints, which could be solved by adoption of local grain handling and banking centers.

## 2.2. Role of LGPBS in Management of Mycotoxins

The scope of the current paper will be limited to activities that directly relate to eradication of mycotoxin contamination. It is proposed that these centers be key points for control of mycotoxins because the problem needs to be tackled at different points in maize value chain [38,41]. Although many agencies have proposed many great ideas to tackle mycotoxin contamination in maize, no single action can individually solve the complex problem, as it involves different points in the maize value chain (Figures 1 and 2). We utilize the publicly available research-based data to propose integrated mycotoxin mitigation strategies to target the roots of the aflatoxin problem tree. The proposed strategies are listed before each point in the problem tree (Table 1). It is envisaged that adoption of these local grain support centers will not only overcome the pre-harvest production constraints, but also replace the currently applied storage systems, which can easily allow for colonization of maize grain by toxigenic fungal species (Figure 3). Further, the localized systems could be designed to overcome the challenges faced with the heaping of grain in large silos at the national reserves, through adoption of modern facilities that provide better grain drying and aeration (Figure 4). By establishing a facility with different support systems to enhance maize production, the potential mycotoxin mitigation strategies are brought together, and the efforts can complement in tackling the challenges at different levels in the value chain (Figure 2). The key sections of the center are expected to interact amongst themselves and with the stakeholders, as illustrated herein (Table 1 and Figure 5).

**Table 1.** Support services and activities to enhance production of safe maize grain in East Africa.

| Point | Activity/Support | Problem Tree Issue | Sections of LGPBS |
|---|---|---|---|
| Pre-harvest | Provision of certified seed of cultivars with desirable traits | Less susceptible maize genotype | -Agronomy advisory team—this would provide farmers with the appropriate information on the best cultivars to grow <br> -Finance credit—this would provide information on monetary support to enhance acquisition of the seed |
| | Provision of farm labour | Improved soil quality | -Agronomy advisory team—inform farmers on the correct tillage method, based on the type of soils in their farms <br> -Finance credit—enhance payment for tillage labour |
| | Input for control of soilborne pests/weeds/other pests and diseases | Improved soil/plant health | -Crop Protection section—to provide information on what pesticides and/herbicides, and the appropriate timing and application rates <br> -Finance credit—to enhance acquisition of the appropriate input |
| | Fertilizer application | Improve soil nutrient content | -Agronomy advisory team—this would provide appropriate information on fertilizer type, rates and timing for application, based on farmers' field conditions <br> -Finance credit—to facilitate purchase of fertilizer |
| | Provision of information on plant spacing | Reduced competition and enhanced plant vigor | -Agronomy advisory team—this would provide the advice to the farmers based on the type of cultivar they grow in their fields |
| | Irrigation | Management of water stress | -Water Harvesting Section—this would support farmers to ensure that the crop gets optimal amount of water <br> -Finance credit—to support water harvesting initiative for the farmers |
| Peri-harvest | Information on proper harvesting equipment | Good harvesting practices | -Agricultural Mechanization team—provide appropriate information on the most sustainable harvesting methods. Could adopt harvesting using special equipment which is provided by the LGPBS |
| | Information on shelling devoid of kernel breakages | Good harvesting practices | - Agricultural Mechanization team—provide appropriate information on the most sustainable harvesting methods. |
| Post-harvest | Information on appropriate grain packaging | Grain handling after harvest | -Postharvest Loss Prevention team—can advise the farmers on how to package the grain. If possible, the LGPBS should take the responsibility of packaging in bags that can allow enough grain drying prior to storage |
| | Collection of grain for delivery to LGPBS | Grain handling after harvest | Postharvest Loss Prevention and Transport sections of the LGPBS to facilitate delivery of the grain to the local reservoir. This should avoid exposure to additional moisture |

**Table 1.** *Cont.*

| Point | Activity/Support | Problem Tree Issue | Sections of LGPBS |
|---|---|---|---|
| Post-harvest | Provision of grain drying services by the LGPBS | Drying to attain optimal grain storage moisture | -Postharvest Loss Prevention team—the LGPBS should have sustainable /inexpensive grain drying methods. To reduce the cost of running the system, modern solar powered driers could be acquired and utilized. They would provide information to farmers about maize cultivars with fast kernel dry-down |
| | Prevention of damage by storage pests | Control of weevil and other storage pests | Postharvest Loss Prevention team—grain could be stored at conditions that do not favour infestation by weevil, moths and rodents. The section could apply recommended pesticides to keep the grain free from damage |

### 2.2.1. Advice on Farm Practices

Farmers' advisory services is a key component in grain production. While this service is a duty of the agricultural extension agents, the specific design of the LGPBS can determine whether these important government workers could take duties within these centers. To provide specific advice to farmers, the centers would involve qualified and experienced personnel on need basis. The services would include how to conduct pre-, peri-, and post-harvest management practices. This would ensure that the crop is produced under optimal conditions.

a.   Good agronomic practices

Proper farm management practices can boost crop vigor and are able to reduce crop stress and the subsequent susceptibility of maize to mycotoxigenic fungi [42]. Crop stress is determined by multiple factors during the growth and development stage. For example, aflatoxin accumulation in maize has been strongly associated with drought, insect damage, and a lack of adequate nitrogen in the soil [8,11,12,43]. The LGPBS centers can play a role of advising farmers on how to adopt sustainable methods to ensure that the crop is produced without water, soil fertility, and biotic stresses, as these would lead to mycotoxin accumulation. Individual stress factors can be managed by adoption of the strategies on case basis.

*Management of soil environment and fertility*: Good soil architecture, aeration, and fertility are components for the maize growing agro-ecologies. Thus, characterization of the physical and chemical aspects of the soil is an important activity prior to a recommendation for crop establishment [44]. While farmers in East Africa have a tradition of growing what they have seen in other farms in the neighborhood, it is imperative that governments should zone crop production activities based on evidence of the prevailing favorable soil conditions for maize production. Certain cropping systems, tillage methods, and application of natural and synthetic fertilizers could be used to adjust and attain the appropriate architecture, but proper expertise advice is necessary [15,44–46]. The LGPBS can play the role of providing advice through which various soil conditions can be overcome to achieve the requirements of maize production. To achieve this, the LGPBS should have established soil analytical capabilities. The LGPBS could also offer advice on the recommended fertilizer application rates to the farmers. Currently, soil testing services are limited to very few institutions, which are located within the major cities of East Africa. Good soil health would translate into better utilization of nutrients by the crop, hence, a higher vigor and less susceptibility to mycotoxin accumulation [8,47].

*Management of weeds, pests, and diseases*: Biotic stresses are parasitic to the crop plants. Weeds compete for water and nutrients or they could attach themselves and deprive the nutrients (e.g., *Striga hermonthica*.) causing up to 85% loss in maize crop [48,49]. Stress due to weed infestation has been strongly associated with aflatoxin and fumonisin contamination in maize [50,51]. Thus, the center can play a key role in advising farmers on accurate timing and rates of application of herbicides, and/or the manual management of the weeds. Insect pests cause damage to field and stored crops. The parasitic field pests deprive maize of the nutrients and water, and their feeding creates infection courts for toxigenic fungal species. Damage of maize by thrips was strongly associated with fumonisin accumulation [52]. Further infestation on mature grain causes breakage and avenues for penetration

and colonization by molds, and, hence, contamination by aflatoxin [15]. Application of insecticides reduces insect damage, hence, enhancing crop vigor. Furthermore, application of fungicides eradicates both true and opportunistic fungal pathogens, some of which are mycotoxigenic. Because many farmers may not know about the proper application of chemicals, there is need for the LGPBS to have expertise who can provide advisory services. The facilities could also be used as points of distribution of products with beneficial microbial organisms to the farms (e.g., the biocontrol products currently being adopted in different countries in Africa) [53].

*Water stress management*: Although there is a lot of climatic data that have been gathered over the years, there is a concern that climate change will bring uncertainty in crop production and food safety. Erratic weather conditions can lead to unexpected drought and floods within the arable regions of East Africa. The magnitude of drought determines whether a certain season would provide any grain to the farmers. Under extreme conditions, there is little or no grain, and the little that is available could be contaminated with aflatoxin [43,52]. In this situation, maize consumers eat what is available, and are likely to be exposed to the damaging toxins. Proper mapping and communication of drought risk to farmers could prevent exposure to damaging toxins. The LGPBS could work with the meteorological departments to provide timely awareness about the weather changes in given locations, and the best crop cultivars/varieties that suits the contemporary seasons. The LGPBS could also provide advice on cropping systems that could conserve moisture and facilitate water harvesting. As a long-term intervention, the centers could work with government agencies to establish inexpensive water harvesting strategies for their schemes.

*Provision and promotion of seed stocks of adapted maize cultivars*: While each of the identified practices for reduction of mycotoxin accumulation through agronomic practices only confer a fraction of the overall effect, identification of an adapted cultivar for individual maize production environments can be important in solving the problem. Adapted maize cultivars possess a cumulative resistance owing to multiple important traits that protect them against the mycotoxin predisposing factors [54]. Although genetic resistance to aflatoxin and fumonisin has not yet been bred into East African maize, the problem can be overcome by growing adapted maize cultivars, as they possess some of the key traits that are associated with reduced contamination [8,55]. Thus, the LGPBS could advise the farmers to grow maize cultivars that are well-adapted to the abiotic and biotic stresses of a given environment [56]. The LGPBS can have a stock of the seed of the adapted and well-performing maize cultivars for provision to farmers during the planting season. Among the traits that have been associated with reduced mycotoxin accumulation in maize are early maturity, tolerances to drought, insect damage, low soil nitrogen, and compactness of the endosperm (flintiness) [8,56,57]. As a long-term intervention strategy, breeders could work with the LGPBS to identify key germplasm for integration of the traits that are correlated with mycotoxin resistance into high adapted and yielding backgrounds using modern breeding methods such as genomic selection [58].

b. Timely and proper management of agronomic practices

Most of the farmers pay for farm labor. In some cases, the farm activities may not be accomplished to the right standards due to lack of enough training of the workers. To overcome the bottlenecks of unskilled labor, LGPBS could establish a pool of trained workers to perform some farm practices for at a fee. The crew would ensure that the major farm activities are performed correctly and within the acceptable timing. For example, timely planting is essential for rain-fed maize because the crop gets the advantage of early establishment, before the onset of other biotic stresses and competitors, and could reduce plant stress and aflatoxin accumulation in maize [59]. Timely control of weeds is essential because it prevents detrimental competition with the crops (Figure 2). Also, pests and diseases must be controlled early enough to avoid epidemics that can lead to extreme plant stress, quality damage, and economic losses [38,60]. Timely harvesting ensures that the activity does not take place when it is rainy, the ears are not over matured, and the kernels have not been broken by weevil, as these would lead to further entry of the toxigenic molds (Figure 2) [12].

The alternative to offering labor for the farmers is to provide advice and training for better pre-, peri-, and post-harvest farm practices. Further, the facilities should provide information about the available products, services, and opportunities throughout the maize value chain. To achieve this, the centers would prepare teaching materials and hold regular training workshops. Regular farm visits to farmers fields by trained personnel is required to ensure proper implementation of the farm practices. In a case where records are given to the farmers, the trainers must ensure that the best possible methods of communication are applied (e.g., most of the farmers are not able to understand the fertilizer and pesticide application rates). Thus, the experts need to know the exact size and the planting density in their farms so that they can recommend the optimum amounts of inputs to be applied in the entire field.

c. Harvesting and post-harvest practices

Peri-harvest activities can play a key role in preventing mycotoxin contamination. For small-scale farmers, harvesting involves cutting off the maize plants and stacking them to remain in the field for several days and/or until they are presumed to be dry [61]. Field stacking does not provide enough aeration for the ears and could lead to colonization of grain by toxigenic fungal strains. Upon drying, maize is de-husked and dropped onto the ground to dry up and then they are collected into bags ready for transport. Dropping of de-husked ears on the ground can expose maize to entry of the spores of toxigenic fungi [62]. Through on-farm demonstrations, LGPBS would serve the role of providing advice on how to handle the maize ears appropriately during harvesting. The LGPBS would provide inexpensive moisture testing equipment at a fee. The facility could also acquire and install multiple modern driers (e.g., EasyDry M500) so that farmers can deliver the grain for a centralized drying [63]. Further, the centers could also provide seeds of maize with fast dry-down. The combined approaches would reduce the grain drying duration. Fast drying would reduce chances of entry of toxigenic fungi, hence, preventing mycotoxin contamination.

Grain shelling should be conducted in a way that minimizes kernel breakage, as these have been associated with increased mycotoxin contamination [15]. The majority of small-scale farmers shell maize using mechanical methods that can cause breakage. To avoid this problem, the LGPBS could establish inexpensive and high throughput mobile on-farm shelling equipment, which could be shared by farmers within the scheme. To ensure that only clean ears are shelled, farmers would be advised to sort and remove ears with apparent moldiness. Removal of moldy ears after harvest was found to significantly reduce aflatoxin contamination in maize [14]. Sorting based on apparent moldiness was also found to reduce the percentage of contaminated samples by more than half [15]. To advance the safety through more efficient sorting, the LGPBS could work with scientists to validate and adopt multi-spectral sorters, as preliminary studies have shown that they can detect and sort both aflatoxin and fumonisin [64].

Upon drying, the grain can be packaged in aerated bags, which can handle a mass that can be easily handled by human operators (e.g., a maximum of 25 kg). The grain can then be transferred to the LGPBS facility for storage. To prevent entry of weevil during storage, regular inspection and fumigation with insecticide should be conducted. To avoid challenges associated with the traditional grain storage systems, the WRS should provide storage systems that include modern facilities (Figure 3). If not prevented, weevils can cause breakage and could introduce opportunistic molds to the stored grain. Recently, there have been many reports of successful hermetic grain storage technologies to prevent weevil damage and subsequent accumulation of mycotoxins. For example, the hermetic plastic and metallic silos, Purdue Improved Crop Storage (PICS), and GrainSafe®bags have been widely recommended for storage of maize and other grains [65]. The LGPBS can evaluate the potential of storage of maize in these improved facilities or adopt the same technologies in larger equipment to accommodate the large volumes of grain from their schemes.

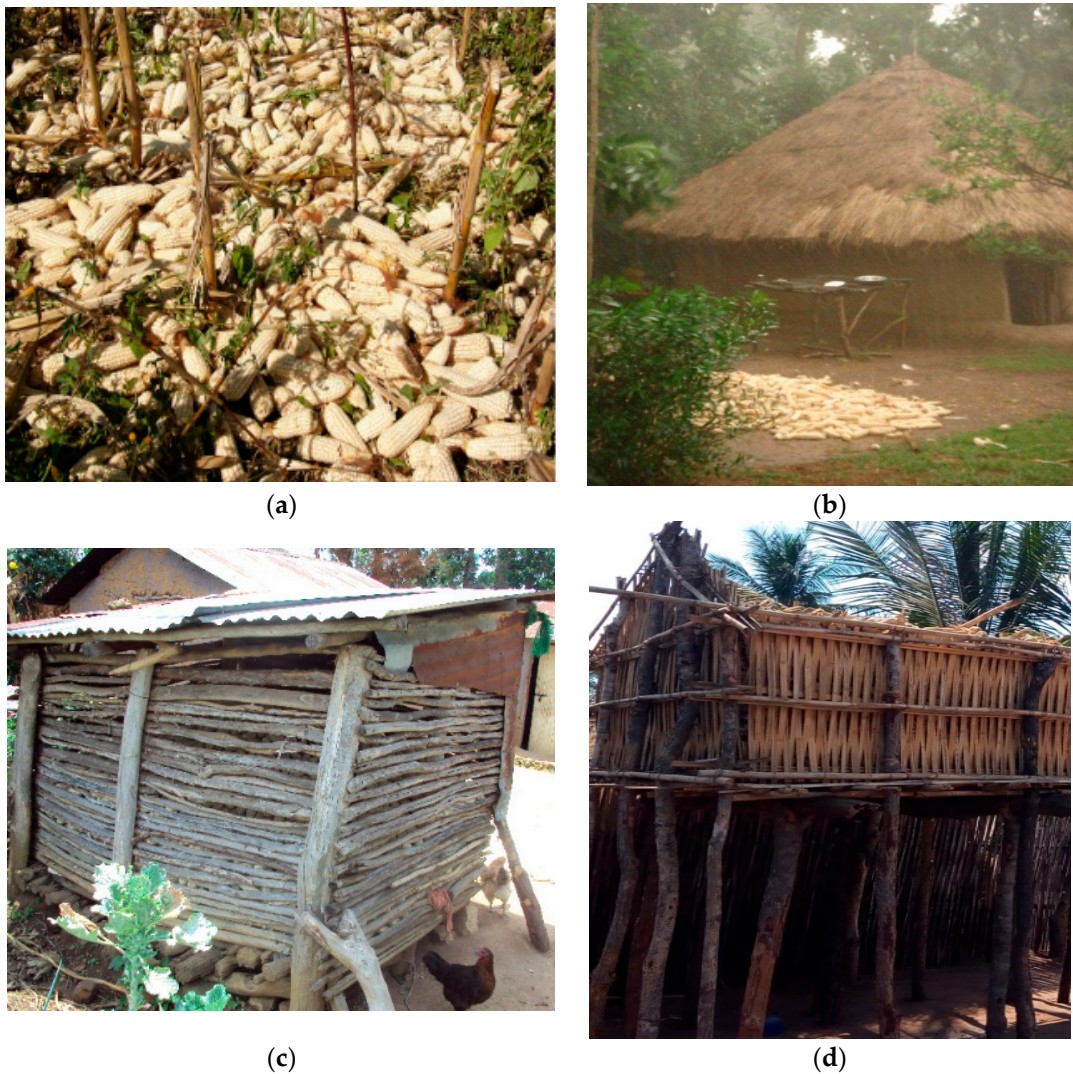

**Figure 3.** Maize handling and storage practices by small-holder farmers in East Africa. (**a**) Maize ears are dropped on the ground during harvesting; (**b**) maize ears are dried on the ground; (**c**,**d**) maize ears are stored in traditional wooden cribs, which are vulnerable to entry of water and rodents. Photos a–c, by S. Mutiga in 2010 within Bungoma district, Kenya and d by A. Mushongi in 2016 within Tunduru district, Tanzania.

d.  Innovations for decontamination and alternative use of contaminated maize

It is anticipated that LGPBS will consult and work with experts at different stages of grain value chains to ensure maximum quality and safety of the produce. One important area of support that would contribute to food safety is research on new technologies. By working with local and international research organizations, these centers can test existing and new technologies. For example, although some interventions were found to reduce mycotoxin contamination at the experimental level, they have not been tested in actual field conditions. A promising technology like spectral sorting has been reported to work in the developing countries and was recently tested at laboratory level in Kenya [64]. Additionally, a recent study showed less aflatoxin in maize kernels of high density, but the potential of density-based sorters has not received adequate support to enhance evaluation [8]. Similarly, addition of diatomaceous earth has been reported to enhance grain drying and to reduce weevil infestation, but this technology has not received enough scientific support to ensure its safety and efficacy in East Africa [66,67]. The facility could also serve as a learning and acquisition center for foreign food preparation practices such as nixtamalization (washing and cooking of maize in an alkaline solution),

as this reduces aflatoxin levels [68]. The facilities could also be used as centers for application of alternative uses of contaminated maize. For example, after sorting, the highly contaminated grains could be used to generate heat energy. The facilities could explore possibilities of utilizing the contaminated grain for production of ethanol. Ethanol production from maize is common in the US [69].

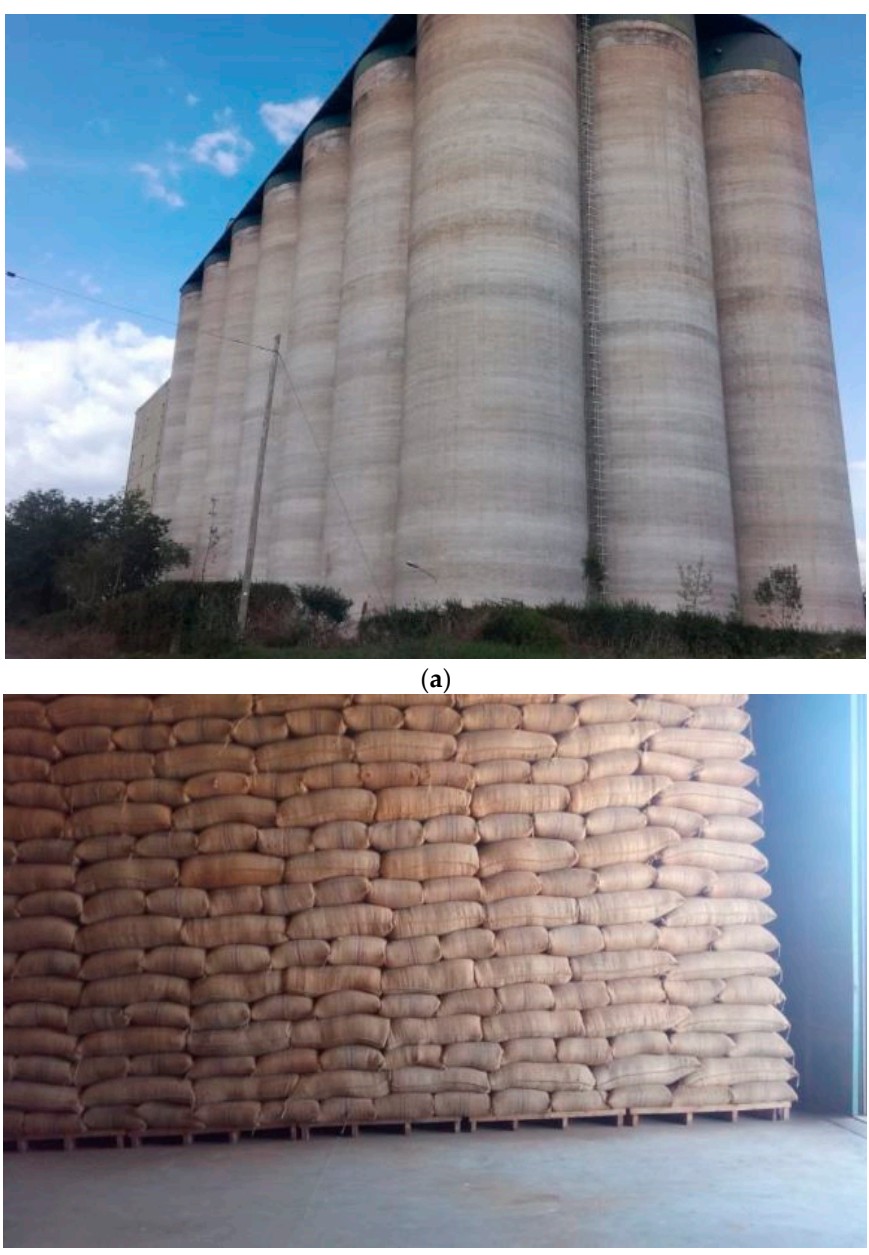

(**a**)

(**b**)

**Figure 4.** Maize packaging and storage within East African national grain reserves. (**a**) a maize storage silo in Nakuru, Kenya. (**b**) maize packaged in typical 90-kg sisal bags and heaped on wooden pallets in a storage silo at Makambako branch of the national food reserve agency, Njombe region, Tanzania. Photos. a, by S. Mutiga, June 2018. b, by A. Mushongi, May 2019.

2.2.2. Farm Input Provision and Related Services

The small-scale farmers of East Africa are faced with financial challenges and may not afford some important farm inputs e.g., fertilizers, herbicides, pesticides [70]. To boost farmers' productivity, provision of inputs on a need basis can be helpful. For most regions of East Africa, input provision are incentives provided by the governments or by donor agencies [70]. However, these interventions are not

sustainable and are faced with inefficiencies due to lack of resources and a general misunderstanding of the requirements for different growing parts of individual countries. To bridge the gap of the lack of actual demands of specific agro-ecologies, the LGPBS would be able to assess the needs and react to farmers in each of the grain-producing areas. Timely provision of herbicides would reduce competition between maize and weeds, and hence, boost crop vigor. Additionally, timely provision of pesticides would ensure that insect pests and diseases are managed before they can achieve economic threshold. The most promising role of the LGPBS would be to stock the inputs and to provide them, together with support services (e.g., advice and labor) to avert plant stress. These incentives would be achieved in a business agreement between the local facility and the farmers. In addition, farmers would benefit from advice on how to apply the inputs appropriately.

### 2.2.3. Sales, Promotions, and Credit Services

LGPBS would be major stakeholders in grain production and would play a role in establishing better external markets for the farmers. This means that they would advertise and promote the grains (and any associated products) from their respective regions. In return, LGPBS would benefit from sales commissions. Furthermore, LGPBS could benefit from interests arising from grain banking systems. On the other hand, farmers would benefit through timely access to advice, inputs and loan services, better quality grain, and reduced exposure to mycotoxins. Individual centers would develop and market their products to the farmers and to other customers outside their geographical areas.

### 2.2.4. Grain Custodians and the Associated Banking Services

LGPBS would aggregate and store grain on behalf of the farmers in a model named the warehousing receipt system [40]. Establishment of the WRS was aimed at banking systems for the grain and would replace the current storage systems, which are characterized by large storage silos (Figure 4).

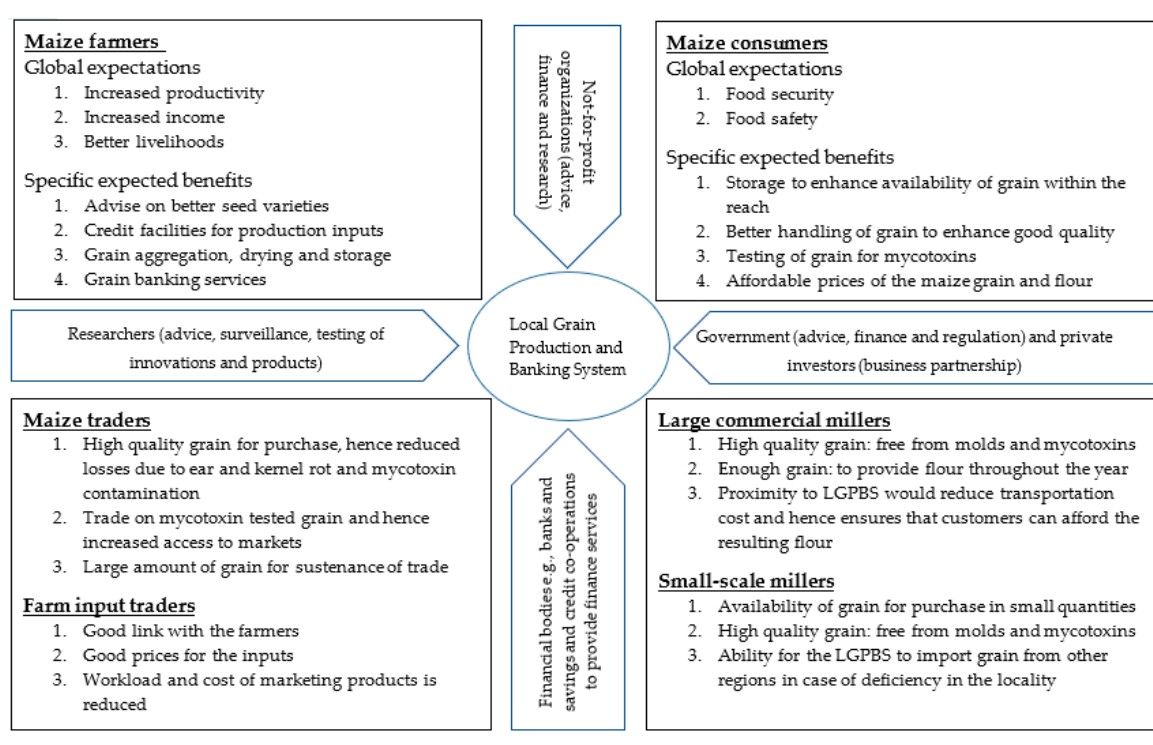

**Figure 5.** Conceptual model of how local grain production and banking system (LGPBS) would be implemented to enhance maize production, safety, and trade in East Africa. The potential supports (e.g., advisory, credit, donations, research, etc.) from different stakeholders to the LGPBS are shown in light green arrows. List of the potential benefits of the LGPBS to millers, traders, farmers, and consumers is contained in white rectangular boxes.

To obtain the custody, the center would develop a system of tracking the transactions for individual farmers, as described by the regulations of individual countries. In the current concept, additional key activities have been described to be implemented at the LGPBS besides grain banking. As a custodian of large quantities of grain from the many farmers, the LGPBS facility would enjoy economies of scale; hence, they can build modern equipment to handle the grain appropriately. As a food safety enhancement measure, grain banking by the LGPBS would ensure that the important produce is not kept under conditions that would favor toxigenic fungi. It would also ensure that the grain is handled by personnel who have a better understanding of the conditions that lead to contamination. Furthermore, because LGPBS are business oriented, there would be a greater sense of responsibility; hence, regular monitoring of the quality would be implemented in the facility. The potential benefits of the proposed LGPBS are summarized in Figure 5.

## 3. The Scope and Limitations of the Concept

While this concept focuses on mitigation of mycotoxin contamination in maize, there are other crops and foods that are vulnerable to contamination [71]. The concept provides a foundation for community-based integrated mycotoxin mitigation strategies, with maize as an example of a popular and a frequently contaminated crop in East Africa [7,15]. Given the high per-capita consumption of maize in East Africa, effective control of mycotoxins in maize would significantly reduce human fatal poisoning and cancer incidences [1,36]. Efforts to enhance production of maize under optimal conditions would not only improve human and livestock health, but also the boost increased production and, hence, food security. If successful, the proposed model could be implemented on other vulnerable crops such as groundnuts.

Production of maize under optimal conditions is hypothetical and relies on human activities and environmental factors. While the proposed concept aims at enhancing optimal maize production conditions to prevent contamination by mycotoxins, participation of farmers would depend on a prior communication of the expected benefits. Improper management of the LGPBS could prohibit participation of farmers. Lack of willingness of farmers to participate in the community-based integrated mycotoxin control scheme could hinder the success of the concept. Furthermore, uncontrollable factors such as the effects of climate change may frustrate the efforts of attaining optimal maize production conditions [72]. To achieve the full benefits of the proposed concept, there is a need for concerted efforts by multiple stakeholders within individual countries and at the regional level. Within a country and its administrative units, policy makers must establish proper tools to engage all the other stakeholders. There must be good communication for the focus, the associated challenges, and the costs.

Cropping systems and farm management differ across agro-ecologies, communities, and for different crops. Furthermore, different crops have different production regimes, including management of abiotic and biotic stresses. In the current concept, we have highlighted the need for establishment of delineated maize production schemes within feasible production agro-ecologies. Partitioning of these schemes could be guided by existing shared traditions in maize production practices, which could be mainly guided by the climatic conditions and existing policies of individual regions.

The culture of self-provisioning of food in most East African communities can hinder the whole community-based mycotoxin mitigation effort [15]. Although some level of self-provisioning is acceptable, a consumer education and policy could be adopted to regulate food items that are historically known to be vulnerable to mycotoxin contamination. This proposed concept prioritizes maize as a food item with a clear history of contamination by the damaging mycotoxins [20,73,74]. Through the LGPBS, consumers would buy or withdraw grain from the central reservoir. For community safety, the reservoir would only circulate grain that has fully been certified not to contain harmful toxins. Because of the need to enhance food safety, it would remain a responsibility of individual LGPBS to establish facilities that are attractive to farmers and to maintain good quality grain for the consumers.

Implementation of the LGPBS concept relies on willingness of investors to participate in supporting the farmers within the model. The principle of enhancing crop production exists but is fragmented

within the value chain. Currently, each player acts in a proprietorship scope by fitting themselves within the points that provide profit. For example, agro-shops sell farm inputs, while banks provide credit facilities based on collaterals. When stakeholders act independently, they may be in favor of profits, risking the common goal of good livelihood of humankind. The challenge in the LGPBS model is to bring these key players under one roof with a focus on enhancing food safety and increased crop productivity. To influence investors into this new business model, there is need for good communication, and provision of enough evidence of the existence of the problem in the value chain.

## 4. Concept Implementation Perspectives

The overall goal of this concept is to have an integrated regional mycotoxin control strategy that originates at community level. The need for regional effort is driven by the fact that maize grain is traded within East Africa, and high levels of contamination in one country would easily spill to the entire region [14]. The need to initiate the effort at community level is similar but has the force of pooling resources together to enjoy economies of scale and, hence, to boost the capacity for farmers to afford farm inputs, labor, and better storage facilities. Furthermore, given the frequent grain trade, mycotoxin contamination within a neighborhood means a health risk to maize consumers, as maize is marketed for consumption at households, institutional levels, and even in big hotels. Exposure to mycotoxins has been associated with increased morbidity, a condition that affects the general health of the community and, hence, reduced income of the people (sick people cannot work effectively) [29]. Thus, effective control of mycotoxins in food and feed value chains would boost the general safety, hence, enhancing income and better livelihoods of the people.

The primary implementer of this model would be the national and local authorities within East Africa. The authorities could play a role in establishment of policy frameworks to ensure that the mycotoxin problem is solved using a community-based approach. To initiate a community-based integrated mycotoxin control system, there is a need for education of maize consumers, policy makers, and regional leaders to combat the problem. A clear description of the complexity of the problem (as shown in Figure 1, case of aflatoxin) and the potential action options would enable the authorities to support adoption of the current proposed model. To enhance operationalization of these centers, there is a need to establish government policy to govern how the LGPBS would operate, with a clear goal of enhancing food safety and productivity. There are some intergovernmental organizations that have played a key role in communications about food safety and mycotoxin regulations. For example, the East African Grain Council (EAGC) and the partnership for aflatoxin control in Africa (PACA) have spearheaded government operations towards eradication of mycotoxins in food and feed. Currently, the East African countries do not have robust mycotoxin surveillance systems, as most people rely on self-provisioned foods. Furthermore, grain handling strategies differ greatly, and the associated policy may not be known to maize growers. Kenya already has some laws in place to initiate the warehouse receipt system, which could support the provisions of the proposed model.

A complete implementation of the LGPBS model would require a lot of infrastructural acquisition and modification. To minimize the costs associated with new infrastructure, existing stakeholders could be amalgamated. Structures that were previously used in the national grain reserves could be acquired and modified appropriately. Furthermore, some stakeholders (those already existing in the maize value chain) could be enlightened and provided with options for the amalgamation process. For example, agro-vets can provide inputs to the centers so that these can be provided to the farmers in each scheme. To initiate the implementation process, pilot models could be implemented with funding from individual governments, non-governmental organizations, and other interested parties. The pilots could be conducted across the country, with each of the key components of the model being tested. To communicate the progress to the stakeholders and potential investors in the subsequent implementation stage, the pilots should measure grain quality and productivity improvements.

## 5. Conclusions

The complex problem of mycotoxin contamination in maize has been a major challenge in enhancing food safety and security in East Africa. Here, we synthesized the publicly available data to develop an aflatoxin problem tree, which contains roots (factors for contamination), the body (the target aflatoxin entry points), and the branches (outcomes on human and livestock). We utilize this tree to propose integrated mycotoxin intervention strategies at the community level for the whole of East Africa in a concept model named local grain production and banking systems (LGPBS). It is envisaged that implementation of this concept could improve grain production, handling, and storage practices, hence, reducing mycotoxin contamination in maize. The concept of LGPBS provides an opportunity to stem the problem of quality loss of grain at all stages of the maize grain value chain.

Establishment and operationalization of the facilities would ensure that stakeholders are cushioned of maize quality and quantity losses caused by abiotic and biotic constraints. Further, the facilities would provide an environment where farmers, who are willing to venture into maize agri-business, are fully supported through provision of inputs, finance, and advisory services. This would, in turn, lead to production of maize under optimal conditions, and hence, reduced mycotoxin contamination, and a subsequent reduced exposure of humans and livestock to the damaging toxins. The post-harvest handling and storage systems would ensure that there is minimal or no grain loss due to ear rot and mycotoxin contamination. This facility, if adopted, would not only lead to enhanced grain productivity, but also improved food safety. An increase in grain production would enhance trade and better livelihoods. On the other hand, enhanced food safety would lead to a healthier population. Therefore, LGPBS provides an excellent concept towards enabling agribusiness, food safety, and food security in East Africa and beyond.

**Author Contributions:** S.K.M conceived the idea of the concept paper. A.A.M and E.K.K reviewed and contributed to the development of all sections of the manuscript.

**Funding:** This research received no external funding.

**Conflicts of Interest:** The authors declare no conflict of interest.

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
