# Peer review of "Enhancing Food Safety through Adoption of Long-Term Technical Advisory, Financial, and Storage Support Services in Maize Growing Areas of East Africa"

_sustainability, doi:10.3390/su11102827_

Round 1

Reviewer 1 Report

The manuscript entitled “Enhancing food safety through adoption of long-term technical advisory, financial and storage support services in maize growing areas of east Africa” presents very important issues, but it requires also some corrections.

 The concept of local grain production and banking system (LGPBS):

- Figure 1 is of poor resolution as well as preparation is messy (uneven arrows)

- Figure 1 – It should be “Aspergillus flavus” instead of “Aspergilus flavus”

- Figure 1 – The arrows in this “problem tree” are sometimes arguable, e.g., “High moisture in store grain” influence the “Entry of highly toxigenic strain of Aspergillus flavus” by “Presence of insect at grain sheds”- It is not always true. Moreover the last branch of this tree (after “contaminated livestock products eaten by human”) must be reconsidered. There is not a simple relationship between mycotoxins in feed and “reduced nutrients absorption”; “physical and mental weakness” etc. Not to mention “unproductive society” and “low income”…

- Table 1 – I think that authors should reconsider the one more action associated with the education actions. Despite the agronomy advisory team efforts in regard to providing “information on the best cultivars to grow; type of soils; type of fertilizer, etc.”; farmers should know and understand the “mycotoxins problem”. 

- Figure 2 needs some corrections (e.g., “Animal feed”) 

- The title of figure 3 needs some corrections (inappropriate underline)

- Figure 5 is of poor resolution

- The limitations section and further perspective are needed. 

Author Response

Thank you for the useful comments. Please find our responses below:

The concept of local grain production and banking system (LGPBS):

-Figure 1 is of poor resolution as well as preparation is messy (uneven arrows):

Response: We have made some revisions on this figure. It must be acknowledged that this tree represents the complex nature of the aflatoxin contamination problem in maize value chain, and could carry more information that we have presented in this figure. We have tried to re-organize it to capture the best representation of the situation. Owing to its complex nature, we have encountered difficulties in fitting it within the main text, and we have attached a separate page with it. We think this figure is important, and should not be treated as a supplement.

-Figure 1 – It should be “Aspergillus flavus” instead of “Aspergilus flavus”

Response: We have fixed this typo. Thanks!

-Figure 1 – The arrows in this “problem tree” are sometimes arguable, e.g., “High moisture in store grain” influence the “Entry of highly toxigenic strain of Aspergillus flavus” by “Presence of insect at grain sheds”- It is not always true. Moreover the last branch of this tree (after “contaminated livestock products eaten by human”) must be reconsidered. There is not a simple relationship between mycotoxins in feed and “reduced nutrients absorption”; “physical and mental weakness” etc. Not to mention “unproductive society” and “low income”…

Response: We acknowledge that the arrows could be arguable. However, the information provided in the figure is based on what is publicly-available (in published literature), and we did our best in the synthesis of the information. For the cases cited, here is the relationship:

1. Moisture vs insects vs fungal colonization: Although high moisture does not automatically bring the storage insects into the storage sheds, they are known drivers for the problem, and it is known obvious that weevil may not attack grain whose moisture content is below 13%. Furthermore, insects are vectors of fungal spores and the spores cannot germinate if the grain is within the acceptable storage moisture. Insects will also break the kernel integrity, and hence enhancing fungal colonization, but only if the grain has the appropriate moisture for the fungus to grow. While the stated phenomena may not be acting alone, the purpose of this figure is to highlight the relationships, and the potential intervention points.

2. Contaminated livestock products eaten by human beings: There is documented high prevalence of metabolites of aflatoxin in livestock products (e.g., milk and eggs), we have provided this citation in the text. While this exposure may not lead to fatal poisoning, it contributes to a chronic health burden. In the revised tree, we have shown that these are joining the consumption of contaminated grain in contributing to the chronic exposure. What these lead to is well documented; increased immunosuppression, increased morbidity etc. These would automatically lead to a society with a health burden and hence reduced productivity, resulting to low income and poverty. In communicating to the policy and the consumers, there is need to highlight these relationships, as they would enable them figure out that there is need to put more effort in ensuring food safety.

-Table 1 – I think that authors should reconsider the one more action associated with the education actions. Despite the agronomy advisory team efforts in regard to providing “information on the best cultivars to grow; type of soils; type of fertilizer, etc.”; farmers should know and understand the “mycotoxins problem”.

Response: We agree that consumers knowledge is important. We have captured this in the text, and we have also added the need to create awareness to other stakeholders, as well.

-Figure 2 needs some corrections (e.g., “Animal feed”)

Response: We have made changes in this figure. Thank you!

-The title of figure 3 needs some corrections (inappropriate underline)

Response: This has been corrected. Thank you!

-Figure 5 is of poor resolution

Response: We have corrected this one.

-The limitations section and further perspective are needed.

Response: We have included these sections. Thanks for your input.

Reviewer 2 Report

The manuscript is a conceptual paper dealing with an important subject of food safety and health risks.

The subject is relevant, however, I have some concerns about the paper:

- From the beginning of the paper, it is somewhat unclear whether the manuscript is a scientific paper or a report on the situation. I would like to see more clearly presented aims/purpose of the study in the introduction. (currently the aim is more clear in the abstract than in the actual manuscript). What is the research questions of the study and what new knowledge does it bring to the "current state of the art".

- The authors say that the paper is conceptual. I would like to see more reflection and discussion on this methodological issue. Who is the main audience for this conceptualization? Or is the main aim to present new ideas and innovations to improve food safety?

- Tables and figures are informative. Table 1 and figure 1 could perhaps be made more coherent or to be included as an appendix.

- I would need to see much more thorough discussion with the current literature and current level of knowledge in the conclusion chapter. What is the main body of knowledge the paper is adding?

Author Response

The subject is relevant, however, I have some concerns about the paper:

- From the beginning of the paper, it is somewhat unclear whether the manuscript is a scientific paper or a report on the situation. I would like to see more clearly presented aims/purpose of the study in the introduction. (currently the aim is more clear in the abstract than in the actual manuscript). What is the research questions of the study and what new knowledge does it bring to the "current state of the art".

Response: We have made changes in the Introduction section to show that we are proposing a community-based integrated approach for mitigation of mycotoxin problem based on known drivers. We have highlighted that we synthesized publicly-available data to show the potential intervention strategies. The proposed model is an extension of the existing warehouse receipt system, which is currently being implemented in some countries. We may not go into listing the research questions, as it is not a research paper.

- The authors say that the paper is conceptual. I would like to see more reflection and discussion on this methodological issue. Who is the main audience for this conceptualization? Or is the main aim to present new ideas and innovations to improve food safety?

Response: We have made changes within the main text to clearly show whom the audience is. The overall goal of this paper is to present a new potential approach in tackling the food safety problem.

- Tables and figures are informative. Table 1 and figure 1 could perhaps be made more coherent or to be included as an appendix.

Response: We have included a paragraph in the introduction section and the discussion to capture the relationship between Table 1 and Figure 1.

- I would need to see much more thorough discussion with the current literature and current level of knowledge in the conclusion chapter. What is the main body of knowledge the paper is adding?

Response: We have added two sections (scope and limitations, and Model implementation perspectives) in the main text. We believe that the revisions will be able to give the paper a better coherency.

Regarding the new knowledge, we believe that the proposed model would enable farmers to improve maize productivity and safety. This has been explained the the text. Thank you very much for the useful comments.

Round 2

Reviewer 2 Report

The manuscript has been improved and now warrants publication in Sustainability

Author Response

Thanks for your support.